# Organic Wastes Augment the Eco-Restoration Potential of Bamboo Species on Fly Ash-Degraded Land: A Field Study

Riya Sawarkar, Adnan Shakeel ⬛, Piyush A. Kokate ⬛ and Lal Singh *⬛

National Environmental Engineering Research Institute, Nagpur 440020, India
* Correspondence: lalsingh@neeri.res.in; Tel.: +91-9404950627

**Abstract:** Rapid industrialization has been a major cause of land degradation and other environmental problems globally. Most energy inputs in industries depend on coal-burning power stations which release various pollutants into the environment. Among these pollutants, fly ash is a concerning pollutant for soil quality, as it occupies a voluminous area of land in India and renders it unproductive. Therefore, this work attempts to evaluate the organic amendment-facilitated bioremediation/phytoremediation of fly ash-degraded land through bamboo plantations under field conditions. Three species of bamboo, *Bambusa balcooa*, *B. tulda*, and *B. bambos*, were planted on fly ash dumpsite soil amended with a combined dose of pressmud and farmyard manure. Results demonstrate that after two years of plantation, all the physicochemical attributes of the degraded land were improved considerably compared to the initial observations. Although all the bamboo species exhibited promising phytoremediation potential, variations were observed in their phytoremediation mechanisms: *B. balcooa* was the most ideal phytostabilizer species for Cu, Zn, and Ni. *B. bambos* was found as an ideal phytostabilizer of Pb and Zn while *B. tulda* was found as a phytoextractor of Cr and Zn. Additionally, all the bamboo species sequestrated atmospheric $CO_2$ considerably, resulting in overall environmental restoration of the degraded area; *B. balcooa* was the most ideal species. Moreover, *B. balcooa* exhibited the highest air pollution tolerance index compared to other species. This study, therefore, recommends that a comprehensive analysis of organic matter-mediated phytoremediation would assist environmental managers to formulate sustainable eco-restoration strategies, ensuring a sustainable solution to land degradation.

**Keywords:** carbon sequestration; farmyard manure; heavy metals; pressmud; soil remediation

## 1. Introduction

Due to rapidly increasing energy demands and industrialization, heavy metal contamination in soils has become a serious environmental hazard on a global scale. India, the second most populated country in the world, consumes about 635 million tonnes of coal to meet its huge energy demands. This bulk process of coal consumption in thermal power stations leaves a voluminous amount of fly ash in the environment, which occupies about 155,676 acres of land in India [1]. This fly ash-occupied land has turned infertile and become a source of toxic heavy metals in agricultural lands and water bodies. These heavy metals can build up in plants through the soil and thus pose a serious threat to livestock and human health. Therefore, their remediation is an immediate need of the hour to check the spread of heavy metals into the food chain and uncontaminated environmental sites [2]. The exploitation of hyperaccumulator plant species through phytoremediation has been a promising approach to cleaning up fly ash-polluted soils. However, to expedite the process of phytoremediation, amendment with organic substances can be an advanced approach to minimize the time duration and maximize the metal uptake coupled with enhanced plant growth of hyperaccumulators on fly ash-affected soils [3].

The use of organic amendments to improve plant growth under heavy metal stress is being primarily focused on internationally with the introduction of the regenerative agriculture concept [4]. In this context, pressmud, which is a byproduct of the sugarcane industry, could be an effective organic amendment to promote the phytoremediation potential of hyperaccumulator plant species in fly ash-contaminated soils. It is a rich source of organic matter (21%), nitrogen (2%), phosphorous (1.95%), potassium (0.5%), and small amounts of calcium and magnesium [5]. Moreover, it contains various microelements, such as manganese, zinc, copper, and iron that are essential for plant growth [6]. India generates about 10–12 million tons of pressmud annually [7]. This pressmud is being used for various purposes, such as chemical extraction, fertilizer production, animal feed, biosorbant, and cement production [8]. However, its application in phytoremediation has not been explored so far. Similarly, FYM has been used as a traditional method to improve soil nutrient status and, ultimately, crop yields in India [9]. Some reports have also reported the use of FYM as an adsorbent of heavy metals [10].

Bamboo is an unexploited plant species for the phytoremediation and eco-restoration of fly ash-induced heavy metal stress in soils [11]. Recent studies have revealed that some bamboo species have a high potential to absorb heavy metals and adapt to metalliferous conditions despite the fact that there have only been a limited number of studies on bamboo for the phytoremediation of fly ash-contaminated soils [12]. In addition to its ability to phytoremediate, it has potential benefits as a timber substitute for forests, an effective $CO_2$ sequester, a plant species to stop soil erosion, and a plant species to reduce particulate air pollution. All these applications make bamboo a multifunctional plant species whose cultivation can play a vital role in the agricultural economy and sustainability [13]. Therefore, this study aims at the screening of different bamboo species for their potential for high biomass production, carbon sequestration, and phytoremediation of fly ash dumpsites. Thus, this work entails evaluating the synergistic effect of sugarcane press mud and FYM as an additive to ameliorate fly ash-polluted soil through changes in soil properties and bamboo biomass.

## 2. Materials and Methods

### 2.1. Study Area

The investigation was done on the fly ash-contaminated cropland near the Koradi thermal power station in Nagpur, Maharashtra, India near the Nilofer landfill (Figure 1). Between 21°14′56″ N 79°55′6″ E and 21°15′58″ N 79°65′8″ E are the location coordinates. The total output from this thermal power plant is 2400 MW. It uses 14,800–15,800 tons of coal every day and generates 2.87 million tons of FA a year [14]. Before planting and two years after planting, researchers assessed the study area's physicochemical characteristics.

### 2.2. Experimental Design and Sampling

Three bamboo cultivars were obtained from the CSIR-NEERI in Nagpur, India and replanted into the treated pits at the experiment site: *Bambusa balcooa* Roxb, *Bambusa tulda* (L.) Voss, and *Bambusa bambos* Riviere and Riviere. The seedlings were planted in 1 × 1 m pits with 4 m × 5 m distance that had been prepared with pressmud (30 kg pit-1) and farmyard manure (10 kg pit-1). Weeding, watering, and pulling out dead seedlings for new ones were done regularly. The seedlings' early growth parameters (height, collar diameter, culms, and litter) were assessed before planting. To assess the levels of heavy metals and the rate of carbon sequestration, plant samples were collected. Each species had three samples taken for examination both before and two years after DAP (days after planting).

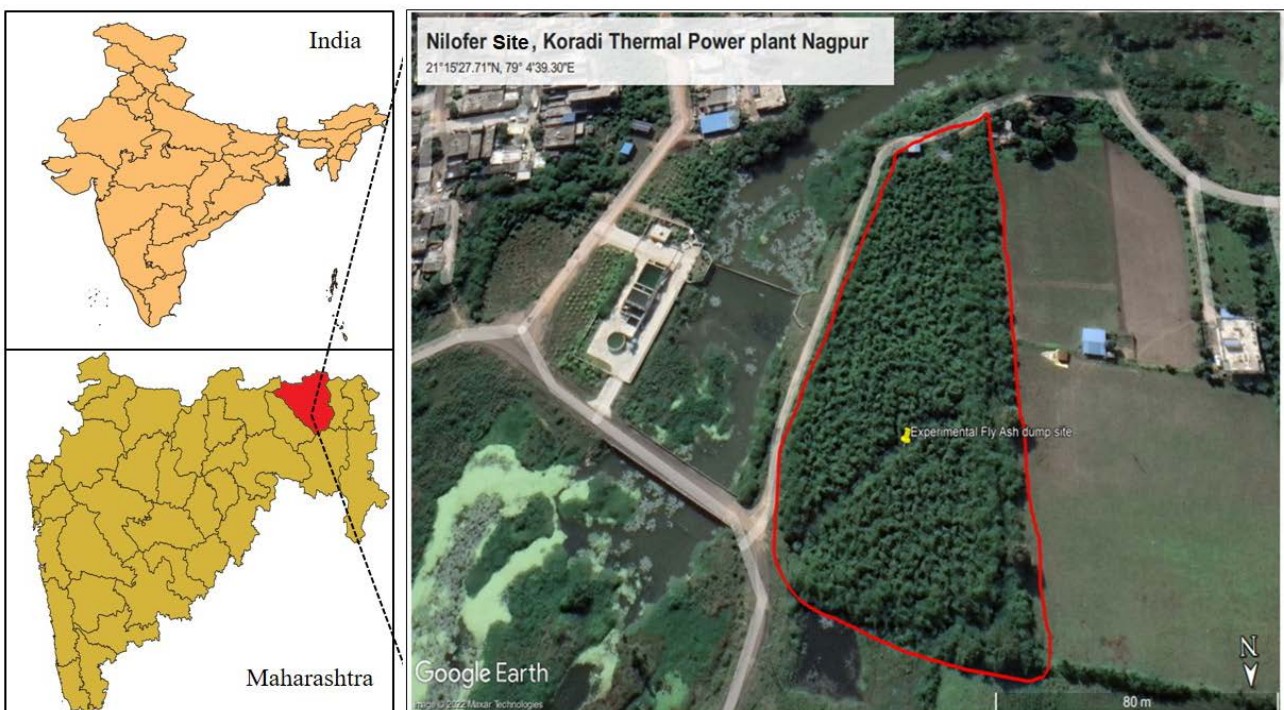

**Figure 1.** Location of experimental fly ash dumpsite, Nilofer Site, Koradi thermal power plant, Nagpur, India.

### 2.3. *Physicochemical Characteristics of Fly Ash Dumpsite Soil*

The gravimetric approach was employed to gauge the soil's moisture content [15]. The moisture content was represented as a percentage (%). By using Keen–Raczkowski box method, evaluations of bulk density (g/cm³), porosity (%), and water holding capacity (%) were performed [16].

#### 2.3.1. pH and EC

To measure the acidity and alkalinity of the soil, HANNA Instruments, Woonsocket, Rhode Island, USA employed a pH meter under the phc101 brand. Using distilled water, 10 g of soil sample was diluted to a volume of 25 mL in a 100 mL beaker (ratio 1:2.5). An electrode was inserted into the suspension after the mixture had been well-stirred, and the pH was then noted. A conductivity meter was used to calculate EC (HI76310, HANNA Instruments and Woonsocket, RI, USA).

#### 2.3.2. Organic Carbon

To calculate organic carbon, the technique of Walkley and Black [17] was employed. One gram of dirt was put into an Erlenmeyer flask after being filtered via a 0.5 mm screen. After rapidly stirring for 1 min, 20 mL of concentrated $H_2SO_4$ and 10 mL of $K_2Cr_2O_7$ were added to the mixture, which was then allowed to sit at room temperature for 30 min. The sample was titrated against 0.5 N ferrous iron (II) ammonium sulfate until the color changed to brown-red, after which, 4 to 5 drops of ferroin indicator were added.

#### 2.3.3. Total and Available Nitrogen

Total N was calculated using the Kjeldhal method [18]. A 0.5 g sample was placed in a digestion tube followed by 1 g digest mixture ($K_2SO_4$: $CuSO_4$) and 10 mL concentrated $H_2SO_4$ and heated to 420 °C at which the sample turned a pale green color. After cooling, the distillation apparatus was loaded with a digestion tube on one side and 4% $H_3BO_3$ along with an indicator on the other side. The distillation machine was set to automatically make up 40% NaOH. The samples were steamed, and the resulting $NH_3$ was collected

in an Erlenmeyer flask containing boric acid. The color of the green distillate changed to the original hue (pink color) after titration with 0.02 N sulfuric acid ($H_2SO_4$). The total nitrogen concentration in the sample was determined using the blank and sample titer (mL) readings. Available nitrogen was estimated using the standardized protocol of Subbaiah and Asija [19].

### 2.3.4. Total and Available P

The total P content was determined by Williams et al.'s [20] approach. A 0.5 g sample was placed in a flask, and then 30 mL digested mixture ($HNO_3$ and $HClO_4$ in the ratio 9:4) was added until the sample became colorless. The above contents were then washed with distilled water, filtered, and diluted to a known volume. Thereafter, 10 mL of ammonium vanadomolybdate was added to the 10 mL of filtered sample in a volumetric flask. The absorption was measured at 440 nm. The phosphate content in the sample was determined using a standard calibration curve. The amount of available P was calculated using the method of Olsen [21].

### 2.3.5. Total and Available K

A flame photometer (Systronics photodiode autoignition, 1027) was used to determine the total K content. In a conical flask (150 mL), 0.5 g of soil sample was taken, to which 30 mL of digested mixture ($HNO_3$: $HClO_4$) was added until the sample became colorless. The abovementioned content was then washed with distilled water, filtered, and distilled water was used to produce it up to a known volume. Available K was determined by following a standard protocol of Schollenberger and Simon [22].

### 2.3.6. Cation Exchange Capacity (CEC)

The Lorenz et al. [23] technique was used to calculate the CEC. A total of 25 g of oven-dried, sieved soil samples were combined with 1 N, $NH_4Oac$ for 60 min, and the solution was then kept at room temperature for the rest of the night. A flask with no soil sample was used as a control. The amount of 0.02 N $H_2SO_4$ added to the collected distillate was titrated until the color became reddish.

### 2.4. Determination of Heavy Metal Accumulation

For the evaluation of heavy metals, soil and plant samples were air-dried, crushed, sieved, and acid-digested. For soil digestion, a 9:4 digestion ratio of $HNO_3$ and $HClO_4$ was employed by following the method of Sun et al. [24]. Following digestion, the samples were exposed to ICP-OES 7000, iCap 7300 DUO, Thermo Fisher, England, and inductive coupled plasma-optical emission spectroscopy for the evaluation of various heavy metals.

### 2.5. Determination of Phytoremediation Indices

BCF and TF were calculated to evaluate the metal phytoremediation capability of the plants. To measure how well plants could uptake heavy metals from the soil, the BCF was determined. TF was responsible for assessing how well plants could transmit heavy metals from the root to the shoot. Phytoremediation indicators were calculated using the method of Mohanty and Patra [25]:

$$BCF = MC_{root}/MC_{fly\ ash}\ (BCF = bioconcentration\ factor;\ MC = metal\ concentration)$$

$$TF = MC_{shoot}/MC_{root}\ (TF = translocation\ factor)$$

### 2.6. Determination of Plant Growth Rate in Bamboo Species

The growth rate of designated species of plant was assessed using a vernier caliper and measuring tape (diameter, height, leaf litter, leaves, and the number of culms). All the bamboo species within the quadrats of 3 m × 3 m were analyzed to measure aboveground biomass (AGB), belowground biomass (BGB), and carbon sequestration rate (CSR) using

the quadrats process. The allometric equations of Pathak et al. [26] were used to calculate ABG, BGB, and CSR:

$$AGB = 0.131 \times DBH^{2.28}$$

$$BGB = 0.26 \times AGB$$

$$CSR = ABG + BGB \times 0.5$$

### 2.7. Determination of Biochemical Attributes in Bamboo Species

Chlorophyll and carotenoid contents were determined through a standard method of Maclachalan and Zalik [27]. The 0.5 g plant leaf samples were macerated in 80% alcohol and after centrifugation; absorbances were recorded at 663 nm, 645 nm, and 510 nm wavelengths.

A fresh leaf sample of 500 mg was homogenized in 20 mL of extraction medium (0.5 g of oxalic acid and 75 mg EDTA in 100 mL of distilled water) to determine ascorbic acid content [28]. A total of 1.0 mL of the homogenate was mixed with 5.0 mL of dichlorophenol indophenols ($20 \text{ g mL}^{-1}$) after centrifugation for 15 min. After shaking, the optical density was determined to be 520 nm using a spectrophotometer (Shimadzu, UV-1800, Kyoto, Japan). To determine the relative water content (RWC), gathered leaf samples were submerged in distilled water overnight and bolted dry, and the weight was taken to determine the turgid weight (TW). These leaves were dried in a 70 °C oven and reweighed to get the dry weight (DW). After that, the relative water content was determined by using the method of Mir et al. [29].

$$RWC = [(FW - DW)/(TW - DW)] \times 100$$

where RWC = relative water content; FW = fresh wt.; DW = dry wt.; TW = turgid water.

### 2.8. Determination of Air Pollution Tolerance Index (APTI) in Bamboo Species

One of the best methods employed for selection of tree species for plantation on roadside and urban settings is based on APTI. It stands for air pollution tolerance index, and the method given by Anake et al. [30] was followed in this study. It is calculated based on four important parameters related to the leaves of these plants. It includes ascorbic acid content, relative water content, total chlorophyll content, and pH of the leaf extract. It is calculated based on the below mentioned formula:

$$APTI = A (T + P) + R/10$$

where A = ascorbic acid; T = total chlorophyll; P = pH; R = relative water content.

$$\text{Total chlorophyll} = ((20.2 \times A645) + (8.02 \times A663)) \times V/1000 \times w$$

where A = absorbance; V = volume of sample; w = weight of sample.

Dust load capacity (DLC) was determined by the formula given by Kumar et al. [31].

$$DLC = W1 - W2/\text{leaf area (mg/cm}^2)$$

where W1 = weight of leaf with dust; W2 = weight of leaf after washing.

### 2.9. Statistical Analysis

One-way analysis of variance (ANOVA) was used to determine the significance ($p$ 0.05) of changes that occurred on the FA-dumped site before and after rejuvenation. Data are given as mean of independent three replicates with SD and P values determined through *T*-test. Principal component analysis was operated through Origin 2021b.

## 3. Results

### 3.1. Impact of FYM and Pressmud on the Physicochemical Attributes of FA Dumpsite

The amendment of the fly ash (FA) dumpsite with the combined application of farmyard manure (FYM) and pressmud improved all its physicochemical attributes after two years of bamboo plantation (Table 1). Among the physical parameters, moisture content was improved by 309%, porosity by 50.86%, water holding capacity (WHC) by 68.80%, and bulk density by 11.26% in comparison to the initial unplanted FA dumpsite. Similarly, the chemical attributes of the dumpsite in terms of pH (1.63%), EC (105.71%), organic carbon (150%), organic matter (239%), total and available N (303.70% and 155.80%), total and available P (104.19% and 312.50%), and total and available K (41.40% and 375%) were enhanced after two years, respectively. The exchangeable cations were significantly enhanced in terms of $Na^+$ (25.50%), $K^+$ (340.60%), $Ca^+$ (405.68%), $Mg^+$ (216%), and, consequently, the cation exchange capacity (112.75%) compared to the initial unplanted FA dumpsite.

**Table 1.** Impact of FYM and pressmud on physicochemical characteristics of fly ash dumpsite after two years of plantation.

| Parameter | Before Plantation | After 2 Years Plantation | *p*-Value ($p \leq 0.05$) |
|---|---|---|---|
| Physical characteristics | | | |
| Moisture (%) | $2.76 \pm 0.03$ | $11.31 \pm 1.67$ | >0.001 |
| Bulk density ($gm/cm^3$) | $1.42 \pm 0.04$ | $1.26 \pm 0.07$ | 0.011 |
| Porosity (%) | $60.2 \pm 5.44$ | $90.82 \pm 4.86$ | >0.001 |
| Water holding capacity (%) | $32.29 \pm 1.05$ | $54.91 \pm 1.45$ | >0.019 |
| Chemical characteristics | | | |
| Ph | $7.95 \pm 0.07$ | $8.08 \pm 0.13$ | 0.066 |
| EC (µS/cm) | $105.77 \pm 3.16$ | $216.6 \pm 13.62$ | >0.001 |
| Organic carbon (%) | $0.18 \pm 0.01$ | $0.45 \pm 0.09$ | 0.011 |
| Organic matter (%) | $0.23 \pm 0.08$ | $0.78 \pm 0.16$ | 0.008 |
| Total N (mg/kg) | $214.67 \pm 13.2$ | $864.89 \pm 45.21$ | >0.001 |
| Available N (mg/kg) | $43.87 \pm 3.96$ | $110.13 \pm 7.92$ | >0.001 |
| Total P (mg/kg) | $143.07 \pm 6.35$ | $292.43 \pm 16.6$ | >0.001 |
| Available P (mg/kg) | $8.5 \pm 1.26$ | $33.1 \pm 5.3$ | >0.001 |
| Total K (mg/kg) | $2836.17 \pm 104.89$ | $4003.17 \pm 179.1$ | >0.001 |
| Available K (mg/kg) | $80.2 \pm 3.78$ | $380.98 \pm 32.84$ | >0.001 |
| Exchangeable cations (meq/100 gm) | | | |
| Na | $1.76 \pm 0.52$ | $2.21 \pm 0.43$ | 0.103 |
| K | $1.97 \pm 0.21$ | $8.68 \pm 1.35$ | >0.001 |
| Ca | $2.64 \pm 0.23$ | $13.35 \pm 1.79$ | >0.001 |
| Mg | $1.93 \pm 0$ | $6.1 \pm 1.2$ | >0.001 |
| CEC | $6.98 \pm 0.92$ | $14.85 \pm 1.21$ | >0.001 |

### 3.2. Impact of FYM and Pressmud on the Heavy Metal Content of FA Dumpsite

The concentration of heavy metals in the fly ash dumpsite before organic amendment and bamboo plantation against the dumpsite after two years of amendment and plantation are given in Table 2. The results showed that there was a considerable reduction in the concentration of total of 10 heavy metals that were detected at the FA dumpsite. The

concentration of all the heavy metals was significantly reduced with maximum reductions in Ba (50%) followed by Cr (49%), Ni (42%), Pb (33%), Co (30%), Mn (29%), Zn (28%), and Cu (25%), respectively, after two years of organic amendment and bamboo plantation.

**Table 2.** Impact of FYM and pressmud on heavy metal concentration of fly ash dumpsite after two years of plantation.

| Elements | Before Plantation | After 2 Years of Plantation | $p$-Value ($p \leq 0.05$) |
|---|---|---|---|
| **Cu** | $95.85 \pm 1.63$ | $72.55 \pm 2.77$ | >0.001 |
| **Cr** | $54.85 \pm 3.46$ | $27.48 \pm 4.6$ | >0.001 |
| **Cd** | ND | ND | ND |
| **Ni** | $70.35 \pm 3.46$ | $40.31 \pm 1.15$ | >0.001 |
| **Mn** | $1296.1 \pm 16.97$ | $908.49 \pm 47.86$ | >0.001 |
| **Zn** | $96.8 \pm 0.57$ | $69.38 \pm 4.89$ | >0.001 |
| **Pb** | $9.13 \pm 0.32$ | $6.06 \pm 0.43$ | >0.001 |
| **Co** | $45.25 \pm 3.25$ | $31.33 \pm 3.58$ | >0.003 |
| **Ag** | ND | ND | ND |
| **Ba** | $470.78 \pm 8.03$ | $232.49 \pm 18.72$ | >0.001 |

ND: Not detected; values are given in mg/kg.

*3.3. Impact of FYM and Pressmud on the Bioconcentration and Translocation Factor of Different Bamboo Species Grown of FA Dumpsite*

The influence of organic waste amendment on the phytoremediation potential of the different bamboo species is given in Table 3. The results clearly demonstrated that *B. bambos* accumulated the highest concentration of Pb in-root (BCF = 1.5) followed by Cu (BCF = 1.34), Zn (BCF = 1.26), Cr (BCF = 0.93), Ni (BCF 0.88), Ba (BCF 0.63), Co (BCF = 0.41), and Mn (BCF 0.12). Similarly, *B. balcooa* accumulated the highest concentration of Ni in-root (BCF = 1.47) followed by Pb (BCF = 1.28), Cu (BCF = 1.11), Zn (BCF = 1.11), Ba (BCF = 1.5), Cr (BCF = 0.66), Co (BCF = 0.18), and Mn (BCF = 0.14). Likewise, *B. tulda* accumulated the highest concentration of Pb in-root (BCF = 1.33) followed by Cu (BCF of 1.01), Cr (BCF = 0.91), Ni (BCF = 0.65), Ba (BCF = 0.39), Co (BCF = 0.35), Zn (BCF = 0.34), and Mn (BCF = 0.10). After accumulation in the roots, all three bamboo species translocated the various heavy metals into their shoots. The translocation factor of *B. bambos* was highest for Ni (TF = 0.60) followed by Cr (TF = 0.57), Cu (TF = 0.34), Zn (TF = 0.24), Ba (TF = 0.14), Mn (TF = 0.11), Pb (TF = 0.02), and Co (TF = 0.01). For *B. balcooa*, translocation was observed to be highest for Cr (TF = 0.51) followed by Cu (TF = 0.40), Zn (TF = 0.30), Ni (TF = 0.19), Mn (TF = 0.19), Mn (TF = 0.19), Ba (TF = 0.06), and Pb (TF = 0.01). In the case of *B. tulda*, translocation was found to be at the maximum for Cr (TF = 1.13), Zn (TF = 1.05), Ni (TF = 0.88), Cu (TF = 0.54), Ba (TF = 0.47), Mn (TF = 0.26), Pb (TF = 0.02), and Co (TF = 0.02), respectively.

**Table 3.** Impact of FYM and pressmud on BCF and TF of bamboo species on FA dumpsite after two years of plantation.

| Plant Samples | | Heavy Metals | | | | | | | | | |
|---|---|---|---|---|---|---|---|---|---|---|---|
| | | Cu | Cr | Cd | Ni | Mn | Zn | Pb | Co | Ag | Ba |
| *B. balcooa* | BCF | 1.11 | 0.66 | 0.00 | 1.47 | 0.14 | 1.11 | 1.28 | 0.18 | 0.00 | 0.71 |
| | TF | 0.40 | 0.51 | 0.00 | 0.19 | 0.19 | 0.30 | 0.01 | 0.00 | 0.00 | 0.06 |
| *B. bambos* | BCF | 1.34 | 0.93 | 0.00 | 0.88 | 0.12 | 1.26 | 1.54 | 0.41 | 0.00 | 0.63 |
| | TF | 0.34 | 0.57 | 0.00 | 0.60 | 0.11 | 0.24 | 0.02 | 0.01 | 0.00 | 0.14 |
| *B. tulda* | BCF | 1.01 | 0.91 | 0.00 | 0.65 | 0.10 | 0.34 | 1.33 | 0.35 | 0.00 | 0.39 |
| | TF | 0.54 | 1.13 | 0.00 | 0.88 | 0.26 | 1.05 | 0.02 | 0.02 | 0.00 | 0.47 |

### 3.4. Impact of FYM and Pressmud on the Plant Biomass and Carbon Sequestration Rate of Different Bamboo Species Grown on FA Dumpsite

The amendment of the FA dumpsite with the combined application of FYM and pressmud significantly enhanced the plant growth parameters of the selected bamboo species (Figure 2). Among the three bamboo species, *B. balcooa* showed maximum improvement in growth in terms of height (328.65%), diameter (408.63%), culms (900%), and litter (1328%). This was followed by *B. bambos* with improvements in height (326.62%), diameter (77.86%), culms (800%), and litter (2664%) whereas *B. tulda* showed height (318.59%), diameter (62.17%), culms (600%), and litter (1532%).

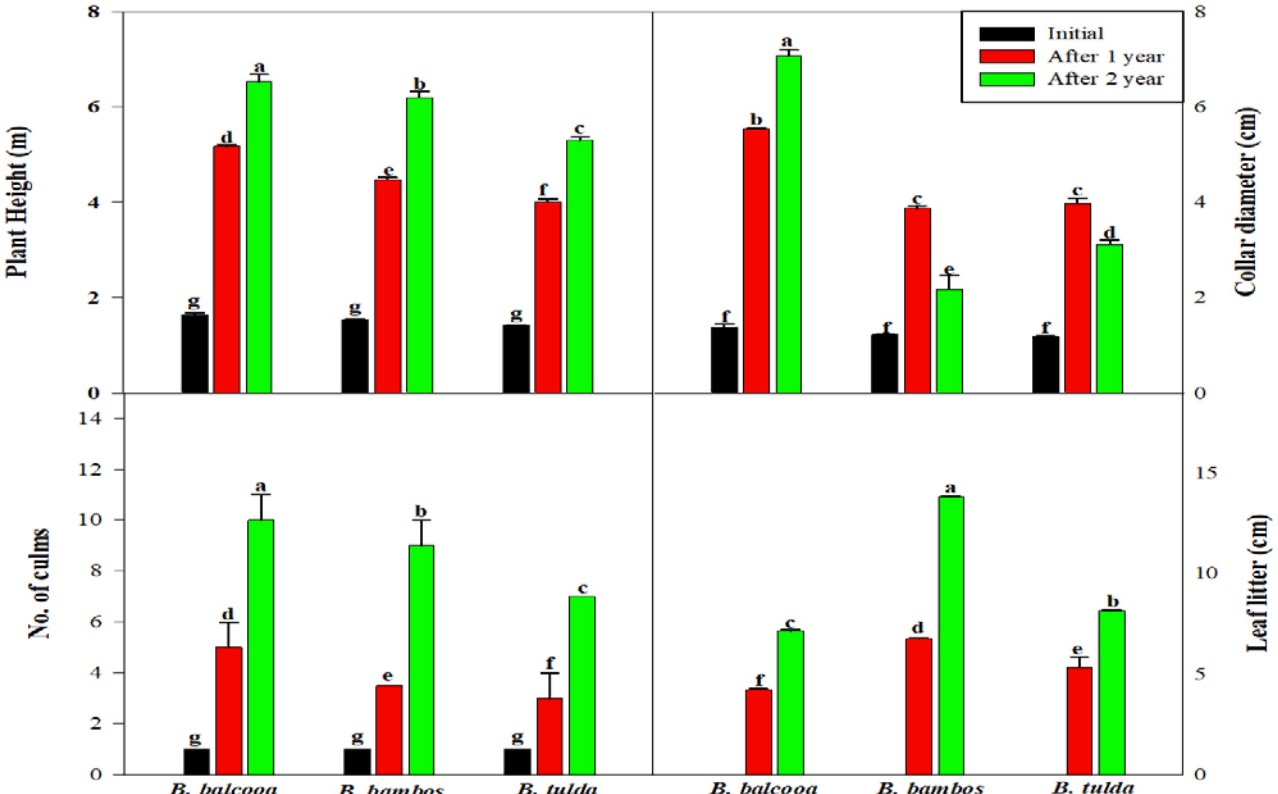

**Figure 2.** Impact of FYM and pressmud on the growth attributes of bamboo species grown on FA dumpsite. Different letters over column bars indicate significant differences while as same letters indicate non-significant differences.

The plant biomass and rates of carbon sequestration in all three bamboo species in terms of aboveground biomass (AGB) and belowground biomass (BGB) were evaluated before and after two years of plantation (Figure 3). The results demonstrate that the maximum improvement in the plant biomasses was observed in *B. balcooa* in terms of AGB (3968.45%) and BGB (3968.02%). Likewise, in *B. bambos*, AGB and BGB were enhanced by 218.72% and 218.71%, respectively. In *B. tulda*, AGB and BGB were improved by 818.62% and 818.78%. The carbon sequestration rate (CSR) of different bamboo species is given in Figure 3. Results revealed that, after two years of bamboo establishment on fly ash dumpsite, CSR was increased significantly in all three bamboo species as *B. balcooa* > *B. bambos* > *B. tulda*. A remarkable improvement in the development of bamboo plant biomass after two years can be seen clearly in Figure 4.

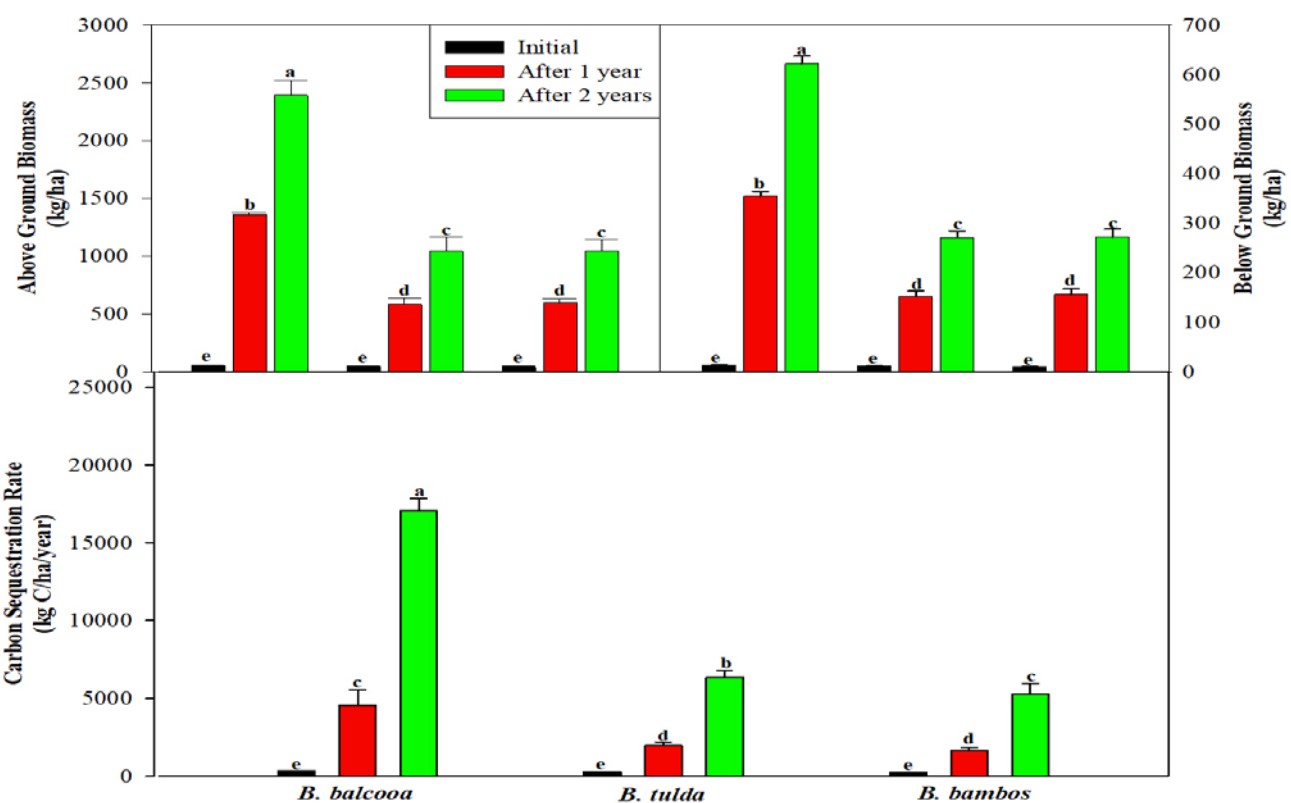

**Figure 3.** Impact of FYM and pressmud on plant biomass and carbon sequestration rate of bamboo species grown on FA dumpsite. Different letters over column bars indicate significant differences while as same letters indicate non-significant differences.

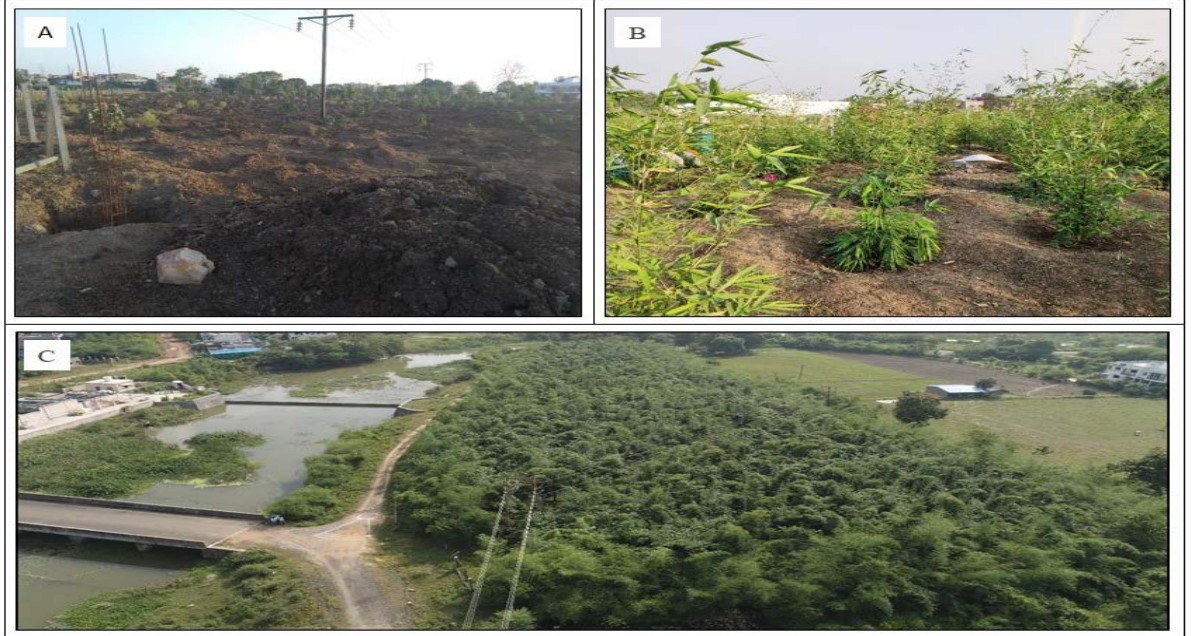

**Figure 4.** Changes in bamboo plant biomass: initial degraded soil (**A**); first-year plantation (**B**); after two years of plantation (**C**).

### 3.5. Impact of FYM and Pressmud on the Photosynthetic Pigments of Different Bamboo Species Grown on FA Dumpsite

The photosynthetic pigments of all three bamboo species *B. balcooa*, *B. bambos,* and *B. tulda* were evaluated during plantation and after two years of plantation (Figure 5). Results showed that *B. balcooa* had the highest improvements in Chl-a (96.77%), Chl-b (80.95%), total chlorophyll (90.38%), and carotenoid content (47.36). This was followed by *B. bambos* with improvements in Chl-a (96.55%), Chl-b, (105.82%), total chlorophyll (100%), and carotenoids (57.14%). The least improvements were observed in *B. tulda* with Chla-a (76.66%), Chl-b (78.94%), total Chl (77.55%), and carotenoids (31.25%). This signifies that *B. balcooa* showed the highest improvement in photosynthetic pigments after amendment with a combined dose of FYM and pressmud.

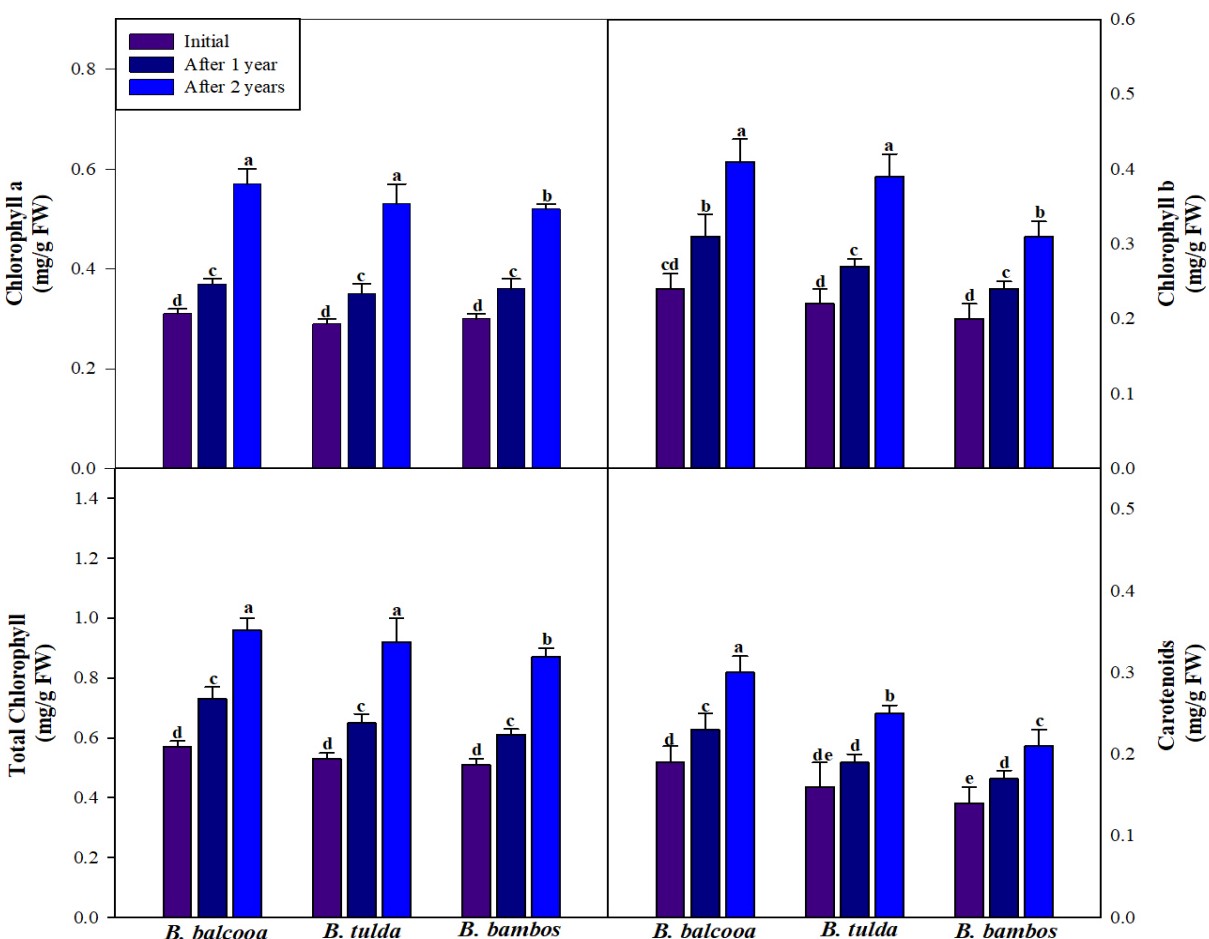

**Figure 5.** Impact of FYM and pressmud on the chlorophyll a, chlorophyll b, and carotenoid content of bamboo species on FA dumpsite. Different letters over column bars indicate significant differences while as same letters indicate non-significant differences.

### 3.6. Impact of FYM and Pressmud on APTI of Different Bamboo Species Grown on FA Dumpsite

After two years of plantation, all three species of bamboo were analyzed for their APTI capacity (Table 4). The results demonstrated that *B. balcooa* has the maximum APTI components in terms of ascorbic acid (8.12 ±0.79), pH (7.33 ± 0.12), RWC (77.59 ± 2.51), and, consequently, the APTI (13.7). Its dust load capacity (0.91 ± 0.02) was also the maximum compared to the other bamboo species. Likewise, for *B. bambos*, the values were ascorbic acid (7.37 ± 0.36), pH (6.2 ± 0.1), RWC (74.07 ± 6.42), APTI (12.66), and dust load capacity (0.47 ± 0.01) while as *B. tulda* showed the least values with ascorbic acid (4.95 ± 0.09), pH (5.97 ± 0.06), RWC (67.02 ± 7.65), APTI (10.09) and dust retention capacity (0.32 ± 0.01). Principal component analysis revealed a significant positive correlation in *B. balcooa* among

the stress tolerance parameters, including, pH, ascorbic acid, total chlorophyll, relative water content, dust load capacity, APTI, and carbon sequestration rates (Figure 6). The biplot showed more than 70% variance in two PCA components, and hence significant variations existed between *B. balcooa* and the other two bamboo species. PCA results are further strengthened by Table 4, which clearly shows that *B. balcooa* has the maximum values for all the stress parameters.

**Table 4.** Air pollution tolerance index (ATPI) of bamboo species after two years on fly ash dumpsite. Different letters as superscripts indicate significant differences while as same letters indicate non-significant differences.

| Plant Sample | Ascorbic Acid (mg/100 g) | pH | Relative Water Content (%) | ATPI | Dust Retention Capacity (mg/cm$^2$) |
|---|---|---|---|---|---|
| *B. balcooa* | 8.12 ± 0.79 [a] | 7.33 ± 0.12 [a] | 77.59 ± 2.51 [a] | 13.7 | 0.91 ± 0.02 [a] |
| *B. bambos* | 7.37 ± 0.36 [a] | 6.2 ± 0.1 [b] | 74.07 ± 6.42 [a] | 12.66 | 0.47 ± 0.01 [b] |
| *B. tulda* | 4.95 ± 0.09 [b] | 5.97 ± 0.06 [c] | 67.02 ± 7.65 [b] | 10.09 | 0.32 ± 0.01 [c] |

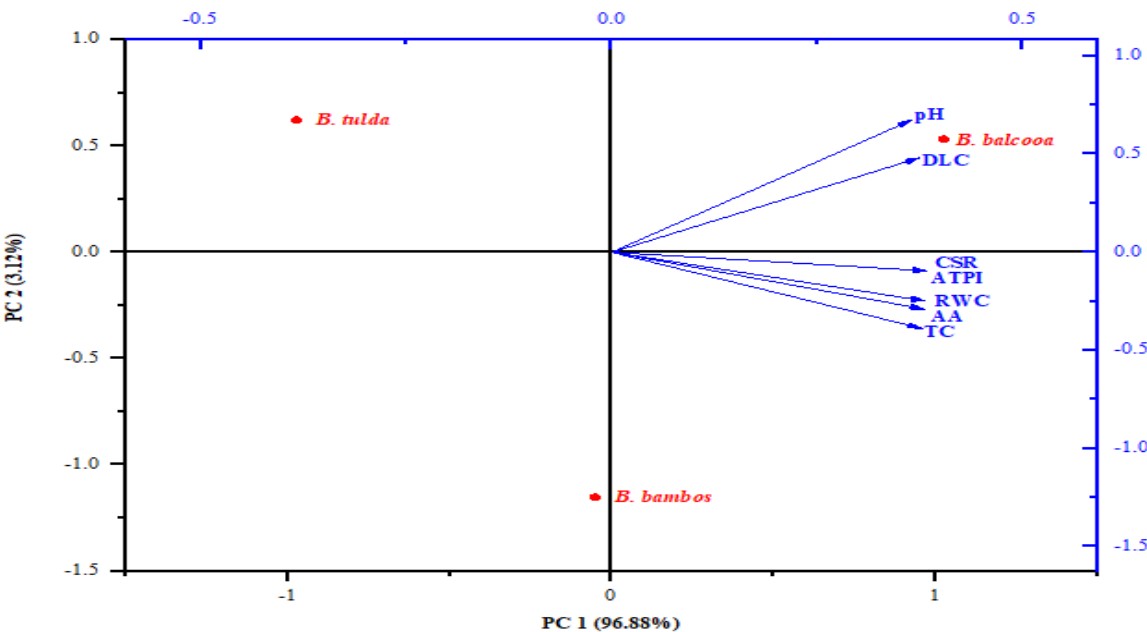

**Figure 6.** Principal component analysis of different stress tolerance parameters of bamboo species, APTI = air pollution tolerance index; DLC = dust load capacity; CSR = carbon sequestration rate; RWC = relative water content; AA = ascorbic acid; TC = total chlorophyll.

## 4. Discussion

Impact of FYM and Pressmud on the Photosynthetic Pigments of Different Bamboo Species Grown on FA Dumpsite.

The use of organic substances in remediation and restoration processes is gaining interest globally to replace chemical methods. Various findings have suggested that organic matter reduced the bioavailability of contaminants in soil and deter their drainage into water bodies, and thus can be employed to clean up or stabilize polluted soils [32,33]. This study evaluates the application of two combined organic substrates (FYM + press-mud) for their potential eco-restoration of a FA dumpsite. Initially, the physicochemical characteristics of the dumpsite were unfavorable to support any plant growth. However, after its amendment with the combined organic substrates (FYM + pressmud), all the physicochemical attributes were enhanced significantly to support the development of bamboo species on the FA dumpsite (Table 1). The physical characteristics of the soil were

also improved considerably after two years of bamboo plantation. The improvements in porosity, WHC, and bulk density can be attributed to the addition of FYM + pressmud, which caused gluing of soil particles with more stable soil aggregates, a decrease in the no. of micropores, and an increase in the volume of macropores in the soil through microbial growth activation [34]. There was a remarkable improvement in the organic matter and nutrient composition of the FA dumpsite after two years of bamboo plantation. This is in correspondence with the richness of organic matter and essential plant nutrients in the FYM and pressmud [35]. Further, the deposition and decomposition of litter due to the bamboo plantation could have contributed to the improvement in organic matter at the FA dumpsite after two years of plantation (Figure 2). Similar findings have been reported by [36]. The improvement in cation exchange capacity after two years of bamboo plantation occurred possibly due to the addition of $Na^+$, $K^+$, $Mg^+$, and $Ca^+$ ions through the addition of the organic substrate (FYM + pressmud) into the FA dumpsite.

This study finds that the simultaneous application of pressmud and FYM remarkably augmented the heavy metal accumulation and phytoremediation of the FA dumpsite (Table 2). Though bamboo is an excellent plant species regarding its phytostabilization capacity, the amendment of organic wastes further enhances its phytoremediation indices. This possibly occurs due to the improvement in soil properties, which consequently increases plant growth metabolism and triggers a high rate of nutrient and metal uptake from the soil [37]. This study corroborates with the findings of [38] which reported that FYM enhanced the phytoextraction potential of maize on metal-contaminated land. Likewise, pressmud has been reported to improve plant growth and bioconcentrations of different metal ions from contaminated soil [39]. Various reports have demonstrated that the concentration of phytoavailable heavy metals relies on the content of organic matter in the soil [40,41]. Organic amendments improve the bioavailability of heavy metals for plant uptake by regulating the microbial community in the rhizosphere of plants [42]. All these studies adhere to the findings of this study and suggest that the application of organic amendments can enhance the phytoremediation capacity of plants under contaminated conditions.

In addition to phytoremediation, this work evaluated the influence of FYM and pressmud on the plant biomass of three bamboo species viz.: *B. balcooa*, *B. bambos*, and *B. tulda* (Figure 3). The combined application of organic waste amendment significantly improved the AGB and BGB of bamboo species. This improvement in plant biomass occurred due to the increased availability of NPK through organic amendment [43]. Moreover, the enrichment in organic carbon through organic waste amendment improved the beneficial plant rhizosphere microbiota and, consequently, the plant biomass [44]. Most importantly, the carbon sequestration rate (CSR) of all the selected bamboo species was improved remarkably with the amendment of FYM and pressmsud (Figure 3). In correspondence with our results, [45] also reported that the addition of FYM significantly enhanced the carbon sequestration rate of rice and wheat.

The air pollution tolerance index (APTI) is an important indicator to evaluate the ecological efficiency of plants under polluted conditions. This study finds that all the bamboo species after amendment with FYM and pressmud showed promising APTI indices. Among the three bamboo species, *B. balcooa* had the highest APTI followed by *B. tulda* and *B. bambos* (Table 4). APTI is based on four components, including, pH, relative water content (RWC), ascorbic acid, and chlorophyll content [46]. All these four components play an important role to maintain the normal functioning of plants under stress conditions [47]. Leaf pH is a vital indicator of plant sensitivity toward atmospheric pollution [48]. Under acidic atmospheric pollution, leaf pH tends to fall, which hinders the photosynthetic efficiency of plants and consequently reduces plant growth. Therefore, tolerant plant species maintain their leaf pH at around 8.8 to withstand acidic conditions [49]. In this study, *B. balcooa* maintained its leaf pH within this range and thus, exhibited high photosynthetic pigments and biomass. Another important physiological component is the RWC, which determines the water status in plants under stress conditions [50]. A high value of RWC signifies the importance of plants under drought conditions. Atmospheric pollution reduces

the transpiration rate, which consequently results in minimal water uptake and low RWC. Tolerant plant species have an RWC within the range of 58 to 73% [50]. Therefore, *B. balcooa*, whose RWC falls well in this range, showed maximum APTI and plant biomass. Ascorbic acid is an important antioxidant vitamin to scavenge the oxidative burst induced in plants under stress conditions [51]. Therefore, tolerant plant species have a higher concentration of ascorbic acid as compared to sensitive plants [52]. In this study, *B. balcooa* had the highest ascorbic acid content and thus maximum tolerance compared to the other bamboo species. Chlorophyll is an important parameter to determine the APTI in plants [53]. The presence of acidic and alkaline pollutants can cause the degradation of stomata through stomata blockage and the formation of phaeophytin [54]. Therefore, tolerant plant species have high chlorophyll content to maintain the required photosynthetic processes under stress conditions [55]. All these four components demonstrate that among the three bamboo species, *B. balcooa* is the most efficient species due to its high APTI, maximum plant biomass, and carbon sequestration rate after two years on the fly ash dumpsite.

## 5. Conclusions

This work evaluates two organic waste amendments, viz., pressmud and FYM, for their potential of improving the phytoremediation and eco-restoration potential of three bamboo species, including *B. balcooa*, *B. bambos*, and *B. tulda*, on fly ash dumpsite soil. Results demonstrated that organic waste amendments ameliorated the soil health indicators of degraded soil significantly after two years through the bamboo-assisted phytoremediation process. Further, this study finds that *B. balcooa* is an excellent plant species with reliable APTI, CSR, DRC, and particulate pollution tolerance. Therefore, this work recommends that among the three bamboo species, *B. balcooa* has a great potential to grow on fly ash-degraded soils, and therefore, this bamboo species should be planted around thermal power plants and other industries for sustainable carbon sequestration and eco-restoration purposes.

**Author Contributions:** R.S.: Experimental work, analysis, and manuscript writing; A.S.: Formatting, statistical analysis, and manuscript writing; P.A.K.: experimental design; L.S.: Review and supervision. All authors have read and agreed to the published version of the manuscript.

**Funding:** This research received no external funding.

**Institutional Review Board Statement:** This study was approved by the research committee of CSIR-NEERI institution with approval KRC No.: CSIR-NEERI/KRC/2022 /DEC/EBGD/2.

**Informed Consent Statement:** Not applicable.

**Data Availability Statement:** Not applicable.

**Acknowledgments:** Authors are very thankful to the Director, CSIR NEERI for proving the facilities to conduct this work. Ethical review and approval is not applicable to this study.

**Conflicts of Interest:** The authors declare no conflict of interest.

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
