# Peer review of "Organic Wastes Augment the Eco-Restoration Potential of Bamboo Species on Fly Ash-Degraded Land: A Field Study"

_sustainability, doi:10.3390/su15010755_

Round 1

Reviewer 1 Report

The Manuscript [sustainability-2124247] entitled (Organic Wastes Augment the Eco-restoration Potential of Bamboo Species on Fly Ash Degraded Land: A Field Study) evaluate organic amendment facilitated bioremediation/phytoremediation of fly ash degraded land through bamboo plantation under field conditions Where B. balcooa exhibited the highest air pollution tolerance index as compared to other species. This study recommends that comprehensive analysis of organic matter mediated phytoremediation would assist environmental managers to formulate sustainable eco-restoration strategies ensuring sustainable solution to land degradation.

Generally, the experiments are well designed and explained. The manuscript has good results and written very well.

Comments

1-      Line 74-75: transfer this objective to the end of introduction.

2-      Line 124: [the technique of Walkley and Black (17) was employed] instead of [the [17] technique was employed]

3-      Line 131: [according to Kjeldahl [18]] instead of [using the [18]], the same in line 158, 179, 205

4-      Figure 1 showing the location map of site should contain international geographical coordinates so that it becomes easy for readers to find it.

5-      Line 214. In determination of Air pollution tolerance Index (APTI), the formula for calculation is total chlorophyll is not given. Please add the formula used.

6-      Table 1 and 2: where the statistical analysis of these tables. It should be analyzed with T-test, then, add T values, df. and P.

7-      Line 268: arrange words those are in the table [3.3. Impact of …..]

8-      In page 9: Figure 2, 3 and 4: add these figures after their mention directly

9-      Table 4: authors should add df, F, and P values in the table or in the text.

Author Response

Thanks for providing your valuable scientific suggestions regarding the manuscript. All the suggested changes have been made to the manuscript and the revised manuscript contains track changes accordingly.

  1. Line 74-75: transfer this objective to the end of introduction.

Response: These lines have been transferred to the suggestion section.

  1. Line 124: [the technique of Walkley and Black (17) was employed] instead of [the [17] technique was employed]

Response: The citation and method has been revised as suggested.

  1. Line 131: [according to Kjeldahl [18]] instead of [using the [18]], the same in line 158, 179, 205

Response: All the methods with required citation style has been updated as suggested.

  1. Figure 1 showing the location map of site should contain international geographical coordinates so that it becomes easy for readers to find it.

Response: Figure 1 has been updated with international coordinates.

  1. Line 214. In determination of Air pollution tolerance Index (APTI), the formula for calculation is total chlorophyll is not given. Please add the formula used.

Response: Formula for total chlorophyll has been added as suggested.

  1. Table 1 and 2: where the statistical analysis of these tables. It should be analyzed with T-test, then, add T values, df. and P

Response: Statistical analysis has been done through T test and significant P value has been added to table 1 and table 2 as suggested.

  1. Line 268: arrange words those are in the table [3.3. Impact of …..]

Response: Words have been changed as suggested.

  1. In page 9: Figure 2, 3 and 4: add these figures after their mention directly

Response: Figures have been placed directly after their mention as suggested.

  1. Table 4: authors should add df, F, and P values in the table or in the text.

    Response: The APTI is given only after two years as initially there was no plantation. Significance has been added through one-way ANOVA-DMRT test.

Reviewer 2 Report

I am attaching a word file as my comments and suggestions for authors before its final publication.

Author Response

Thanks for the systematic and timely review of the manuscript. All the suggestions have been included in the revised manuscript. The revised manuscript contains track changes in the revised sections

  1. The methodology section should be updated, as the protocols used are not fully explained. would like to mention the method used for biochemical estimation of chlorophyll and RWC must be given in brief with appropriate reference.

Response: The methodology for chlorophyll and RWC has been updated as suggested.

  1. I suggest the authors to add a separate section of conclusion and future prospects. The important outcome of the Ms must be discussed in the conclusion section.

Response: A separate section of conclusion, which cotains the outcome and future recommendations have been added to manuscript as suggested.

  1. The authors have made an interesting representation of the data through PCA analysis, but unfortunately, I can’t found a detailed explanation of it. The PCA results should be discussed more briefly.

Response: A detailed explanation have been added to the PCA plot as suggested.

Specific comments

Line 15 : FYM? Use the full form

Response: It has been expanded as suggested.

Line 161:N H2SO4 ?? Correct it

Response: Revised as suggested.

Line 165: of HNO3 and HClO4? Whole Ms need to be crosschecked to correct the

chemical formulas used.

Response: All molecular formulas have been updated as suggested.

ml or mL ? A similar abbreviation pattern must be adopted throughout the MS.

Response: A uniform unit has been maintained in the revised MS as suggested.

Table 2 : Cu ( 95.85±1.63 72.55±2.77)?? is it % or mg or kg? please mention the

units in the table.

Response: A footnote has been added to the table 2 with unit mg/kg.

Table 3: I suggest the authors to use ND (Not detected ) in place of 00.

Response: 00 has been replaced with ND as suggested.

  1. balcooa in terms of AGB (3968.45%) and BGB (3968.02%)??? Check it

Thanks

Response: The percentage improvements in biomass were observed because bamboo is a very fast-growing plant species and grows even by centimetres a day. Therefore, remarkable enhancement in ABG and BGB was observed as compared to initial seedling biomass,